# Prognostic Evaluation of Metastatic Castration Resistant Prostate Cancer and Neuroendocrine Prostate Cancer with [^68^Ga]Ga DOTATATE PET-CT

**DOI:** 10.3390/cancers14246039

**Published:** 2022-12-08

**Authors:** Mehmet Asim Bilen, Akinyemi Akintayo, Yuan Liu, Olayinka Abiodun-Ojo, Omer Kucuk, Bradley C. Carthon, David M. Schuster, Ephraim E. Parent

**Affiliations:** 1Department of Hematology and Medical Oncology, Emory University School of Medicine, Atlanta, GA 30322, USA; 2Winship Cancer Institute, Emory University, Atlanta, GA 30322, USA; 3Department of Radiology and Imaging Sciences, Emory University School of Medicine, Atlanta, GA 30322, USA; 4Department of Radiology, Mayo Clinic, Jacksonville, FL 32224, USA

**Keywords:** DOTATATE, PET, prostate cancer, neuroendocrine prostate cancer

## Abstract

**Simple Summary:**

Prostate cancer is the most common cancer in men and, along with the aggressive neuroendocrine variant of prostate cancer, is known to express high levels of the somatostatin receptor. This study explored the feasibility of using the somatostatin binding radiopharmaceutical, [^68^Ga]Ga-DOTATATE PET/CT, to identify metastatic lesions in 17 men with known metastatic castrate resistant prostate cancer or neuroendocrine prostate cancer. All patients demonstrated [^68^Ga]Ga-DOTATATE avid lesions corresponding to sites of disease as identified by CT. Additionally, we retrospectively correlated the degree of [^68^Ga]Ga-DOTATATE to treatment response and found that men with marked [^68^Ga]Ga-DOTATATE uptake in their metastatic deposits had significantly worse outcomes compared to those with moderate or mild [^68^Ga]Ga-DOTATATE uptake. Conversely, men with only mild [^68^Ga]Ga-DOTATATE uptake in their metastatic deposits had a favorable prognostic outcome.

**Abstract:**

Objectives: Prostate cancer is well known to express high levels of somatostatin receptors and preliminary data suggests that PET imaging with the somatostatin analog, [^68^Ga]Ga-DOTATATE, may allow for whole body staging of patients with metastatic castration resistant prostate cancer (mCRPC) and neuroendocrine prostate cancer (NePC). This study explores the utility of [^68^Ga]Ga-DOTATATE PET-CT to identify metastatic deposits in men with mCRPC and NePC and prognosticate disease progression. Methods: [^68^Ga]Ga-DOTATATE PET-CT was performed in 17 patients with mCRPC and of those, 2/17 had NePC. A semiquantitative analysis with standardized uptake values (SUV) (e.g., SUVmax, SUVmean) was performed for each metastatic lesion and reference background tissues. [^68^Ga]Ga-DOTATATE uptake in metastatic deposits was further classified as: mild (less than liver), moderate (up to liver average), or marked (greater than liver). Serial prostate-specific antigen measurements and patient survival were followed up to 3 years after PET imaging to assess response to standard of care treatment. Results: All patients had at least one metastatic lesion with identifiable [^68^Ga]Ga-DOTATATE uptake. Marked [^68^Ga]Ga-DOTATATE uptake was found in 7/17 patients, including both NePC patients, and all were non-responders to systemic therapy and died within the follow up period, with a mean time to death of 8.1 months. Three patients had mild [^68^Ga]Ga-DOTATATE uptake, and all were responders to systemic therapy and were alive 36 months after [^68^Ga]Ga-DOTATATE imaging. Conclusions: [^68^Ga]Ga-DOTATATE is able to identify mCRPC and NePC metastatic deposits, and lesions with [^68^Ga]Ga-DOTATATE uptake > liver may portend poor outcomes in patients with mCRPC.

## 1. Introduction

Prostate cancer (PCa) is the most common cancer among men in the United States [1]. Androgen deprivation therapy (ADT) is the mainstay of treatment for advanced prostate cancer; however, most patients will eventually develop androgen-refractory disease, or metastatic castration resistant prostate cancer (mCRPC). While small cell neuroendocrine prostate carcinoma is rare, a subset of patients previously diagnosed with prostate adenocarcinoma may develop neuroendocrine features after ADT. Transformed neuroendocrine prostate cancer (NePC) has molecular and genetic changes making them resistant to traditional mCRPC therapies, including androgen receptor (AR) targeted therapies [2], and it has been hypothesized that some of the difficulty in treating patients with mCRPC may in fact be due to neuroendocrine differentiation [3]. The reported prevalence of NePC has varied between studies, possibly due to limitations in targeted tissue sampling and heterogeneous disease penetrance. For example, in a study of 450 patients with mCRPC by Perez et al., only 3 patients demonstrated NePC differentiation [4], whereas Jimenez et al. found NePC differentiation in 92/183 metastatic biopsies from 79 of 157 patients with mCRPC [5].

Somatostatin, a neuropeptide that suppresses prostate growth and neovascularization by inducing cell-cycle arrest and apoptosis, is highly expressed in NePC cells [6,7]. Somatostatin receptors, specifically SSTR2, have been shown to be upregulated in PCa and NePC [8,9]. [^68^Ga]Ga-DOTATATE (NETSpot^®^) is a FDA-approved PET radiotracer with high affinity for SSTR2 [8]. Preliminary case reports suggest that [^68^Ga]Ga-DOTA labeled somatostatin analogs may have high sensitivity in identifying sites of mCRPC in addition to NePC [10,11,12,13]. In a recent study involving 12 patients with mCRPC, all patients had at least 1 blastic neuroendocrine metastasis with increased radiotracer uptake [14]. Patients with multiple bone metastases also had significantly higher SUVmax when compared to patients with few metastases.

The goal of this study was to determine the feasibility of identifying mCRPC lesions, including patients with NePC, using noninvasive imaging and to assess whether such an early imaging biomarker can predict eventual progression in patients with mCRPC about to start first line treatment (abiraterone acetate or enzalutamide) for castration resistant disease.

We hypothesized that patients with higher levels of [^68^Ga]Ga-DOTATATE uptake on an initial PET scan may have a shorter time to progression while on oral agents (abiraterone acetate or enzalutamide) due to resistance to antiandrogen based therapeutics [15] when compared to those with lower [^68^Ga]Ga-DOTATATE uptake. We assessed for correlations between the degree and intensity of uptake of [^68^Ga]Ga-DOTATATE with subsequent progression of disease, determined by standard of care whole body bone scans [16] and CT/MR imaging [17], as well as clinical parameters.

## 2. Materials and Methods

### 2.1. Patient Population

#### 2.1.1. Subject Recruitment

Patients with biopsy proven PCa were recruited from the Winship Cancer Institute at Emory University from 18 April 2018 to 16 May 2019. All procedures performed in studies involving human participants were in accordance with the ethical standards of the institutional and/or national research committee and with the 1964 Helsinki declaration and its later amendments or comparable ethical standards. The recruitment protocol was approved by the Institutional Review Board (IRB) and complied with the Health Insurance Portability and Accountability Act (HIPAA). The data was collected as part of a feasibility trial for neuroendocrine prostate cancer imaging under the IRB title ‘Molecular Imaging with [^68^Ga]Ga-DOTATATE PET to Investigate Neuroendocrine Differentiation in Prostate Cancer Patients (IRB#99167)’. This study is listed on clinicaltrials.gov (NCT 03448458). The radiotracer was labeled using a provided NETSpot kit (Advanced Accelerator Applications USA, Inc. Millburn, NJ, USA) and generator produce [^68^Ga]Ga (Eckert & Ziegler Radiopharma, Inc. Berlin, Germany). Safety monitoring during the drug infusion was performed, and no adverse events were recorded. Written informed consent was obtained from every study participant. This study did not interfere with standard patient evaluation or delayed therapy.

Male patients with known biopsy proven mCRPC were selected under the inclusion criteria of 18 years of age or older, with skeletal, visceral and/or nodal involvement, and able to undergo [^68^Ga]Ga-DOTATATE PET-CT. Patients were previously treated with a combination of ADT ± antiandrogen therapy or chemotherapy prior to diagnosis of mCRPC and subsequently were treated with either hormonal therapy or chemotherapy after mCRPC status (Table 1). No intervention or tissue analysis was performed after the [^68^Ga]Ga-DOTATATE PET-CT. In total, 17 patients were recruited and received at least one [^68^Ga]Ga-DOTATATE PET-CT. One patient was subsequently found to have an additional metastatic pulmonary squamous cell carcinoma and was excluded from this analysis. These patients had a mean age of 62.8 years (range from 48y–87y) and were included per the inclusion criteria (Table 1). The two patients with biopsy proven NePC received platinum-based chemotherapy after [^68^Ga]Ga-DOTATATE PET-CT. Of the 14 patients without NePC, 7/14 received enzalutamide or abiraterone, and 7/14 received taxane based chemotherapy after [^68^Ga]Ga-DOTATATE PET-CT. The median PSA at time of imaging was 46.8 ng/mL (range: 4.44–1033.78 ng/mL).

#### 2.1.2. Image Acquisition

All patients underwent [^68^Ga]Ga-DOTATATE PET-CT of the whole body 55–70 min after intravenous bolus injection of 200 ± 11 MBq (5.4 ± 0.3 mCi) of [^68^Ga]Ga-DOTATATE. PET-CT images were acquired on a GE Discovery-690 16 slice integrated PET-CT scanner (GE Healthcare) without IV contrast. CT scan of the skull base to proximal thighs (80–120 mA) was utilized for anatomic imaging and correction of emission data. PET images were acquired in 3-min scan time per bed position PET acquisition. Dead time, detector efficiency and scatter corrections were applied using the routines supplied by the manufacturer. The resulting images were quantitatively calibrated with 6 mm isotropic resolution. Images were reconstructed with the iterative technique and reviewed on a MimVista workstation (MIM Software, Version 7.2.1). Reconstruction parameters were VUE point FX with 3 iterations/24 subsets and 6.4 mm filter cutoff, and the reconstructed slice thickness was 3.75 mm.

#### 2.1.3. Image Analysis

[^68^Ga]Ga-DOTATATE PET-CT images were interpreted by a board-certified nuclear radiologist, blinded to details of clinical history (beyond inclusion criteria) and other imaging. Whenever possible, 3-dimensional PET-Edge conformational ROI were used to encompass the entire structure, otherwise conformational ROIs were utilized to record uptake in regions of physiologic and abnormal uptake. Up to 5 representative index lesions in each category were selected as markers of [^68^Ga]Ga-DOTATATE uptake. If five lesions in each category could not be defined in a patient, all demonstrable lesions up to five were utilized. Lesions chosen were independent but may have coincided with index lesions on conventional imaging.

### 2.2. Semiquantitative PET and Visual Analysis

[^68^Ga]Ga-DOTATATE in prostate/bed and extraprostatic sites such as lymph nodes, visceral organs, and the skeleton were quantified using maximum standardized uptake values (SUVmax) and mean standardized uptake values (SUVmean). SUV values and standard bi-dimensional size measurements of lesions were recorded for up to 5 bone and 5 soft tissue lesions in each patient. The SUVmax of pathological findings were compared with accumulation of the radiotracer (SUVmax and SUVmean) in the liver, bone marrow (L3), and blood pool as reference organs. Summation of [^68^Ga]Ga-DOTATATE uptake parameters from all indexed lesions per patient was also recorded. In addition, visual comparison of the most [^68^Ga]Ga-DOTATATE avid lesions was performed as a simple way to stratify the degree of uptake similar to that performed with Krenning scoring for patients with non-prostatic neuroendocrine carcinoma [18]. [^68^Ga]Ga-DOTATATE PET uptake was visually classified with the lesion either having: (A) Mild [^68^Ga]Ga-DOTATATE uptake (less than liver); defined as [^68^Ga]Ga-DOTATATE uptake ≥ blood pool or L3 marrow < average liver (SUVmean) (Figure 1). (B) Moderate [^68^Ga]Ga-DOTATATE uptake (equal to liver); defined as ≥average liver (SUVmean) and ≤liver (SUVmax) (Figure 2). (C) Marked [^68^Ga]Ga-DOTATATE uptake (greater than liver); defined as >liver (SUVmax) (Figure 3). For normal liver and bone marrow, SUVmean was determined by placing a spherical VOI (3.0 cm in diameter) in a representative healthy part of the organ. Whole blood uptake (SUVmean) was measured by placing a spherical VOI (1.0 cm in diameter) in the left ventricle of the heart. Lesions were determined to be equal to liver SUVmean if within 20% of the measured value.

[^68^Ga]Ga-DOTATATE PET-CT uptake in metastatic deposits was correlated with response to subsequent standard of care therapy as identified by: Prostate-specific antigen (PSA) progression or PSA response (defined as a drop of >50%), change in clinical management, clinical progression as defined by their medical oncologist, progression free survival and patient mortality. Following [^68^Ga]Ga-DOTATATE PET-CT, patients were treated according to standard of care per the clinician’s discretion with oral agents (abiraterone acetate or enzalutamide) or other FDA-approved agents. The results of the [^68^Ga]Ga-DOTATATE PET-CT did not influence the clinician’s treatment decisions. Subsequent routine imaging and standard laboratory analysis (e.g., PSA level) were performed according to the clinician’s discretion. Patients were followed in the context of this study (per clinical routine) for at least one-year after [^68^Ga]Ga-DOTATATE PET-CT.

### 2.3. Statistical Analysis

Since this was a pilot study with the main goal to evaluate the feasibility, statistical power was not calculated.

## 3. Results

### Semiquantitative PET and Visual Analysis

All patients had at least one lesion with identifiable [^68^Ga]Ga-DOTATATE uptake as evidenced by abnormally increased focal [^68^Ga]Ga-DOTATATE uptake with CT correlates (e.g., lymphadenopathy, sclerotic osseus lesions). Lesions with marked [^68^Ga]Ga-DOTATATE uptake per visual and semiquantitative analysis were found in 7 of 16 patients analyzed and with a Gleason score range of 7–9. All patients with marked [^68^Ga]Ga-DOTATATE uptake were non-responders to systemic therapy, all of which died within the follow up period, with a mean time to death of 8.1 months (range of 14.4–92.6 weeks). One patient was confirmed to be NePC and another with shown to have small cell neuroendocrine carcinoma and acinar adenocarcinoma on biopsy. Six of the 16 patients were found to have moderate [^68^Ga]Ga-DOTATATE uptake and all had a Gleason score of 9. Of the patients with moderate [^68^Ga]Ga-DOTATATE uptake, four died with a mean time of death of 13.3 months (range of 12.7–89.6 weeks). The two surviving patients with moderate [^68^Ga]Ga-DOTATATE uptake both had an initial PSA response to therapy and no NePC was found. The three remaining patients with mild [^68^Ga]Ga-DOTATATE uptake had a Gleason score range of 7–8. All patients were all still alive up to 36 months after [^68^Ga]Ga-DOTATATE study and all had an initial PSA response to therapy (Figure 4).

On [^68^Ga]Ga-DOTATATE PET, all patients had at least one lesion with a median number of seven lesions per patient. A total of 11/16 patients had bone and visceral/nodal lesions, 4/16 had only bone lesions, and one had only nodal disease. Summed SUVmax was significantly higher in the 2/16 patients with proven NePC compared to the 14/16 patients with non-neuroendocrine mCRPC (99.1 ± 16.5 ng/mL vs. 48.4 ± 40.6 ng/mL; *p* = 0.04). Generally, the patients with the highest summed SUVmax also had the highest lesion SUV and worse outcomes. However, there was no correlation between summed SUVmax and PSA. SUVmean lesion values followed similar trends to the reported SUVmax analysis; however, there was no easily identifiable comparable reference organ (e.g., liver, spleen) and for ease of analysis and reproducibility this analysis is limited to SUVmax. Next generation sequencing data was available from 6/16 patients. Of these, 1 patient without NePC had a BRCA2 mutation and also had the highest summed uptake in this study.

## 4. Conclusions

NePC is underdiagnosed, as biopsy of metastatic lesions after spread to soft tissue and bones, which is required for histologic confirmation, is rarely done [3,19]. Furthermore, the standard biopsy is subject to sampling error both within a lesion itself and on a global basis, and there is no established non-invasive imaging or biochemical marker to identify NePC. Almost all prostate cancers show focal neuroendocrine differentiation, but about 5–10% of patients with PCa, have a large number of clustered NE cells that are detected by chromogranin A immunostaining [20]. In tumors that can be classified as NePC, the neuroendocrine component usually comprises 5–30% of the tumor mass [21,22]. The mechanisms by which neuroendocrine cells influence prostate carcinogenesis are not fully understood. The transdifferentiation process from a typical epithelial-like to a neuroendocrine-like phenotype may be a consequence of the selective pressure induced by treatments that result in decreased androgen levels, stimulation of neuroendocrine and neural factors, and loss of tumor suppressors and genomic stability [23,24].

Definitive criteria defining high, moderate, and mild uptake have not been established. However, extrapolating a simple interpretation criteria based on literature examples [14] is as follows: (A) Mild [^68^Ga]Ga-DOTATATE uptake defined as [^68^Ga]Ga-DOTATATE uptake ≥ blood pool or L3 marrow < liver (SUVmean). (B) Moderate [^68^Ga]Ga-DOTATATE uptake ≥ average liver (SUVmean) and ≤liver (SUVmax). (C) Marked [^68^Ga]Ga-DOTATATE uptake > liver (SUVmax) [14]. We are not aware of any literature that has attempted to correlate [^68^Ga]Ga-DOTATATE uptake to clinical outcomes in patients with prostate cancer. In this study, we showed that higher lesion ^68^Ga DOTATATE uptake compared to physiologic background [^68^Ga]Ga-DOTATATE uptake portends a worse prognosis, regardless of Gleason score, PSA, extent of disease, or systemic therapy. As expected, [^68^Ga]Ga-DOTATATE uptake in lesions has been found to be higher in mCRPC patients with NePC. All patients with [^68^Ga]Ga-DOTATATE uptake greater than liver (SUVmax) died within 2 years of imaging regardless of Gleason score, PSA value, or therapeutic management. This subgroup included the two patients with NePC differentiation. In patients with moderate [^68^Ga]Ga-DOTATATE uptake, these also had a worse prognosis compared to patients with mild [^68^Ga]Ga-DOTATATE uptake, who were all still alive at the conclusion of this study. We included in our study patients with mCRPC who were about to start the first line of treatment (abiraterone acetate or enzalutamide) for castration resistant disease. We believe that this inflection point best balances the early detection of potential neuroendocrine transdifferentiation with imaging, as these patients typically have higher ECOG (Eastern Cooperative Oncology Group) performance status than those further along the continuum of their treatment.

Case reports exist of increased [^68^Ga]Ga-DOTATATE uptake in prostate adenocarcinoma as well as prostate carcinoma with neuroendocrine differentiation [25]. Typically, NePC does not express PSMA, but there are a few case reports of patients with known NePC also demonstrating several foci of increased PSMA PET uptake [12,26]. Prostate adenocarcinoma is known to upregulate somatostatin receptors resulting in [^68^Ga]Ga-DOTATATE PET uptake, which does not imply by itself the transdifferentiation to a neuroendocrine pathway [14]. Benign conditions such as benign prostate hyperplasia have similarly been shown to have upregulated somatostatin and are [^68^Ga]Ga-DOTATATE PET positive [27,28]. Patients with mCRPC and NePC and intense [^68^Ga]Ga-DOTATATE uptake may be amenable to treatment with PRRT ([^177^Lu]Lu-DOTATATE; Lutathera) [29,30]. [^11^C]C-choline and [^68^Ga]Ga-DOTATATE have been shown to have comparable detection rates for mCRPC [31].

There are several major limitations of our study. One major drawback of our study is the relatively small patient population for both mCRPC and NePC, with a total of 16 patients, including two with NePC. Additionally, this small group was heterogeneous in the Gleason grading and maintenance therapy prior to imaging with [^68^Ga]Ga-DOTATATE. All patients were maintained on ADT prior to [^68^Ga]Ga-DOTATATE PET-CT, but some patients were also maintained on oral agents or platinum chemotherapy. However, despite such a small and heterogeneous group of patients and lesions, we were able to identify differences in [^68^Ga]Ga-DOTATATE uptake between the two groups and correlate uptake with outcomes. An additional limitation is that pathological confirmation was not available for most lesions and focal transformation to neuroendocrine differentiation was not able to be systemically evaluated. Moreover, this study was a feasibility study and was not powered to correlate [^68^Ga]Ga-DOTATATE uptake to outcomes or neuroendocrine transdifferentiation. Additionally, there was no correlative molecular imaging performed on these patients such as with [^68^Ga]Ga PMSA-11 (Locametz^®^) or [^18^F]F PSMA DCFPyL (Pylarify^®^). This lack of correlative imaging limits the ability to further characterize potential transformation to neuroendocrine prostate cancer, as these typically have low PSMA uptake. Finally, it is not known if there is an optimal temporal point after beginning systemic therapy to discriminate either neuroendocrine transdifferentiation or clinical outcomes.

[^68^Ga]Ga-DOTATATE PET provides high contrast images that are able to identify both mCRPC and NePC lesions. This study suggests that simple evaluation with SUV_max_ may provide prognostic information to the treating physician and allow them to guide treatment accordingly. Further investigation with larger data sets is needed to confirm these preliminary findings and to further establish optimal [^68^Ga]Ga-DOTATATE PET imaging parameters.

## Figures and Tables

**Figure 1 cancers-14-06039-f001:**
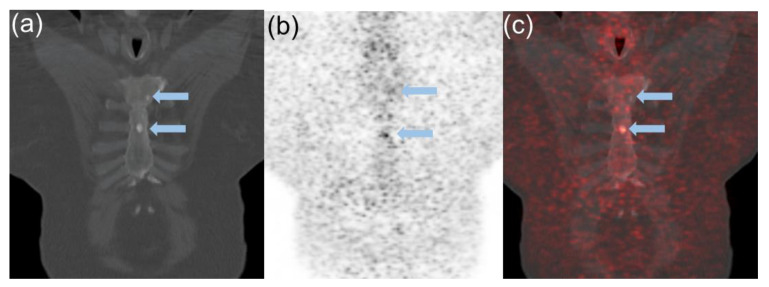
64 year old male diagnosed with Gleason Score 4 + 3 = 7 prostate cancer in 2012. Was previously treated with Bicalutamide (Casodex) and currently maintained on Lupron. At time of [^68^Ga]Ga-DOTATATE PET-CT, patient presented with rising PSA of 11.2 ng/mL with a doubling time of 1.8 months. Coronal images of CT (**a**) [^68^Ga]Ga-DOTATATE PET (**b**) and fused [^68^Ga]Ga-DOTATATE PET-CT (**c**) demonstrate mild [^68^Ga]Ga-DOTATATE uptake in sclerotic lesions (blue arrows; SUVmax 3.1) greater than marrow but less than liver (SUVmax of 17.3). Patient was subsequently placed on Enzalutamide with subsequent PSA response to therapy and survived to the end of study.

**Figure 2 cancers-14-06039-f002:**
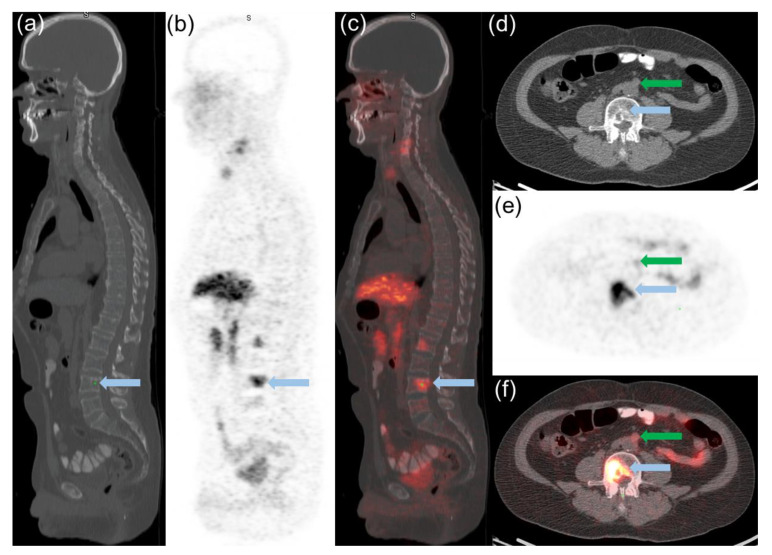
51-year-old man with prostate cancer (Gleason score 9) maintained on ADT since 2011 and placed on enzalutamide four months prior to [^68^Ga]Ga-DOTATATE PET-CT. Selected sagittal CT (**a**) [^68^Ga]Ga-DOTATATE PET (**b**) and fused [^68^Ga]Ga-DOTATATE PET-CT (**c**) images show moderate [^68^Ga]Ga-DOTATATE uptake less than liver (SUVmax 11.8) in several osseus lesions, with the most avid L3 osseus lesion of SUVmax 10.7 (blue arrow). Transaxial CT (**d**) PET (**e**) and fused (**f**) [^68^Ga]Ga-DOTATATE PET-CT views through the L3 vertebral deposit also demonstrates a preaortic lymph node (green arrow) with mild [^68^Ga]Ga-DOTATATE uptake (SUVmax 3.4). Patient was switched to Docetaxel after [^68^Ga]Ga-DOTATATE PET-CT with initial PSA response to therapy but eventually progressed and died 20 months after [^68^Ga]Ga-DOTATATE PET-CT.

**Figure 3 cancers-14-06039-f003:**
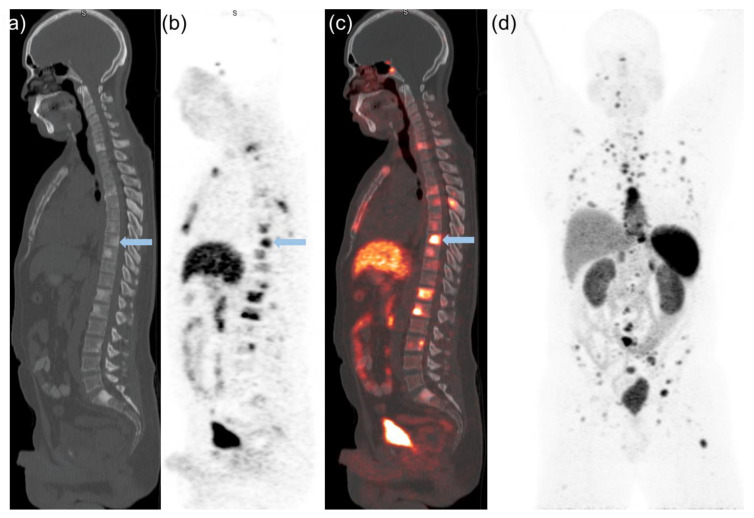
47-year-old man with Gleason score 9, mixed prostate small cell neuroendocrine carcinoma and acinar adenocarcinoma. Patient was stated on ADT and cisplatin/etoposide prior to [^68^Ga]Ga-DOTATATE PET-CT. Selected sagittal CT (**a**) [^68^Ga]Ga-DOTATATE PET (**b**) and fused [^68^Ga]Ga-DOTATATE PET-CT (**c**) images show marked [^68^Ga]Ga-DOTATATE uptake greater than liver (SUVmax of 14.4) in several osseus lesions with the most avid T8 lesion having a SUVmax 20.1 (blue arrow). Anterior view of [^68^Ga]Ga-DOTATATE PET MIP (**d**) demonstrates multiple [^68^Ga]Ga-DOTATATE and nodal metastatic deposits. Patient did not demonstrate a PSA response to therapy and passed away 4 months after [^68^Ga]Ga-DOTATATE PET-CT.

**Figure 4 cancers-14-06039-f004:**
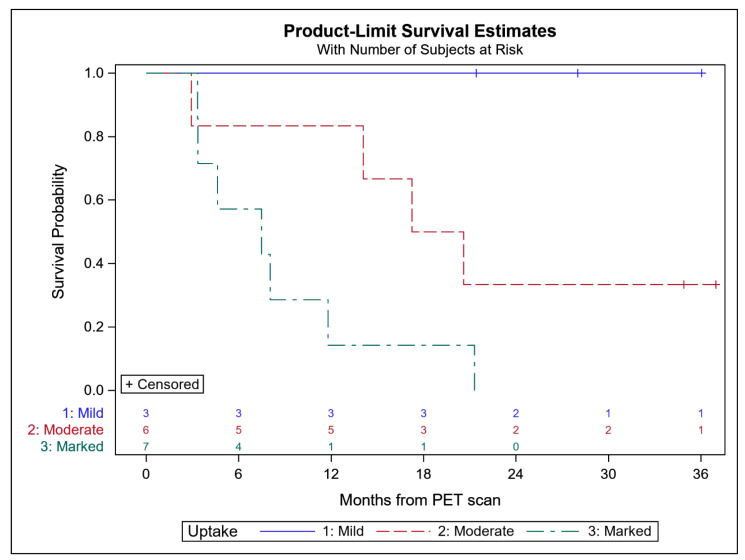
Kaplan-Meier Plots for mCRPC patients with mild, moderate, and marked [^68^Ga]Ga-DOTATATE uptake with a log-rank *p*-value of 0.01.

**Table 1 cancers-14-06039-t001:** Patient demographics.

Gleason Grade	PSA at Time of PET/CT (ng/mL)	Index Lesions with [^68^Ga]Ga DOTATATE Uptake	SUVmax Hottest Lesion	Systemic Treatments Prior to CRPC Diagnosis	Additional Systemic Treatments after CRPC Diagnosis
4	39.23	7	2.1	ADT alone	Enzalutamide
3	11.18	7	3.1	ADT alone	Enzalutamide
2	7.09	3	3.3	ADT alone	Enzalutamide
N/A *	48.08	6	6.2	ADT alone	Docetaxel
5	4.4	2	6.6	ADT alone	Enzalutamide
5	44.54	8	7.3	ADT alone	Docetaxel
5	38.99	9	8.3	ADT+ Docetaxel + anti-androgen therapy	Abiraterone
5	1033.87	7	8.6	ADT+ Abiraterone	cabazitaxol + carboplatin
5	28.76	9	8.9	ADT+ Docetaxel + anti-androgen therapy	Abiraterone
5	31.37	6	10.7	ADT alone	Docetaxel
5	115.45	5	11	ADT+ Docetaxel + anti-androgen therapy	Cabazitaxol
5 ^†^	26.51	8	20.1	ADT + platinium/etoposide	Enzalutamide
4	128	6	20.2	ADT+ Abiraterone	Docetaxel
5	61.86	9	23.1	ADT alone	Enzalutamide
2	88.05	13	27	ADT+ Abiraterone	Docetaxel
N/A ^‡^	<0.01	8	28.5	platinium+ etoposide	Cisplatin + etoposide

* Gleason grade not available. ^†^ Small cell neuroendocrine carcinoma and acinar adenocarcinoma. ^‡^ Small cell carcinoma.

## Data Availability

Data supporting the reported results can be obtained from the corresponding author.

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
