# Peer review of "Prognostic Evaluation of Metastatic Castration Resistant Prostate Cancer and Neuroendocrine Prostate Cancer with [68Ga]Ga DOTATATE PET-CT"

_cancers, 2022, doi:10.3390/cancers14246039_

Round 1
Reviewer 1 Report
In this manuscript, the authors explored the utility of 68Ga-DOTATATE PET-CT to identify metastatic deposits in men with mCRPC and NePC and prognosticate disease progression. It can give a new way to explore the prostate cancer.There are several concerns to the authors:
1. The style of Table should be in three-line.
2. For the caption of Fig. 3, what is yo?
3. The authors should analyze the P value of the components in Figure 4.
Reviewer 2 Report
Dear Authors,
The goal of this study is very interesting, important, and current in the field of nuclear medicine. In general, the manuscript is well designed, the studies have been properly conducted (especially that, the Authors are aware limitations of this study e.g. small and heterogeneous patients group), and the obtained results are interesting. Overall this paper could be accepted after minor revision noted below.
1. What was the reason why the results obtained from [68Ga]Ga-DOTA-TATE did not influence the clinician’s treatment decisions?
2. I would like to suggest add images of prostate cancer tissue from histological examination compared to [68Ga]Ga-DOTA-TATE uptake
3. Ga-68 DOTATATE is incorrect description and should be corrected. Please change to [68Ga]Ga-DOTA-TATE. Please follow the EANM GUIDANCE; please see H.H. Coenen et al. Nuclear Medicine and Biology 55 (2017) v–xi; https://www.eanm.org/content-eanm/uploads/2019/12/EANM_GUIDANCE-_TRACER_NOMENCLATURE.pdf and correct other radioconjugates described in your manuscript e.g. 177Lu-DOTATATE
4. Please check the sentence line 255 - 258 – some letters/words are missing; „In patients with moderate 68Ga DOTATATE uptake, these also ha approved…..at the conclusion of this study.”
5. Correct some typos e.g. line 36 „… cancer; however,…” line 48; „…. with mCRPC.[5].”; line 274 „…MCRPC”
6. Please format the references style as required by the Journal
